# Plasma Turbulence in the Near-Sun and Near-Earth Solar Wind: A Comparison via Observation-Driven 2D Hybrid Simulations

**Luca Franci** [1,2,*], **Emanuele Papini** [2], **Daniele Del Sarto** [3], **Petr Hellinger** [4], **David Burgess** [1], **Lorenzo Matteini** [5], **Simone Landi** [6] and **Victor Montagud-Camps** [4]

1   School of Physical and Chemical Sciences, Queen Mary University of London, London E1 4NS, UK
2   National Institute for Astrophysics (INAF)—Institute for Space Astrophysics and Planetology (IAPS), 00133 Rome, Italy
3   Institut Jean Lamour, CNRS UMR 7198, University of Lorraine, BP 50840, CEDEX, 54011 Nancy, France
4   Astronomical Institute, Czech Academy of Sciences, 141 00 Prague, Czech Republic
5   Department of Physics, Imperial College London, London SW7 2BX, UK
6   Dipartimento di Fisica e Astronomia, Largo Enrico Fermi 2, 50125 Firenze, Italy
*   Correspondence: l.franci@qmul.ac.uk

**Abstract:** We analyse two high-resolution 2D hybrid simulations of plasma turbulence with observation-driven initial conditions that are representative of the near-Sun and the near-Earth solar wind. The former employs values of some fundamental parameters that have been measured by the Parker Solar Probe at 0.17 au from the Sun, while, in the latter, they are set to average values typically observed at 1 au. We compare the spatial and spectral properties of the magnetic, ion velocity, and density fluctuations, as well as the time evolution of magnetic reconnection events that occur spontaneously as the result of the development of turbulence. Despite some differences due to the different plasma conditions, some key features are observed in both simulations: elongated ion-scale Alfvénic structures form in between vortices whenever the orientation of the magnetic field lines is the same, i.e., magnetic reconnection via the formation of an X point cannot occur; the magnetic and density fluctuations at sub-ion scales are governed by force balance; the magnetic compressibility at sub-ion scales is compatible with isotropic magnetic field components; the characteristic time of the formation of current sheets is the eddy turnover at the energy injection scale, while the characteristic time for their disruption via reconnection is compatible with the Alfvén time of the background turbulence.

**Keywords:** space plasmas; plasma astrophysics; solar wind; interplanetary turbulence

## 1. Introduction

The solar wind is a turbulent, weakly collisional, magnetised plasma flowing from the solar corona and filling the whole heliosphere, e.g., [1,2]. Despite decades of observational, theoretical, and numerical studies, solar wind heating and acceleration still represents one of the most fundamental open questions in space plasma physics. The solar wind turbulence properties and their evolution with the radial distance from the Sun have been investigated using in-situ spacecraft measurements at different heliocentric distances, e.g., [3–5]. These studies have shown that protons follow neither the adiabatic prediction nor the double-adiabatic one for a spherical expansion and need to be (anisotropically) heated [6–8]. Different mechanisms, such as turbulence, kinetic instabilities, and magnetic reconnection, typically have an anisotropic effect on the particle thermal energetics. All these processes are also strongly linked to each other, and disentangling their relative contributions is not straightforward and requires a combined effort of observations and modelling. Thanks to radial alignments between the Parker solar probe (PSP) and Solar Orbiter, it is now possible to investigate the evolution of the same solar wind parcel during its expansion starting closer to the Sun than ever before, up to 0.1 au. Recent results have shown that the near-Sun solar wind is highly Alfvénic and characterised by less

developed turbulence, and that it evolves with the radial distance from the Sun towards fully developed and intermittent turbulence, e.g., [9].

High-resolution numerical simulations of turbulent plasmas represent an invaluable tool for interpreting spacecraft data and disentangling the different heating and acceleration mechanisms, e.g., [10–36]. On the one hand, they provide key additional information that are not available from single-spacecraft measurements (e.g., electromagnetic fields and particles' moments in tens or hundreds of million grid points simultaneously). On the other hand, they can be used as controlled experiments to test specific plasma conditions and the relative contributions of the different processes under varying plasma parameters. In this regard, observation-driven hybrid kinetic simulations, employing initial conditions built from average plasma parameters measured by spacecraft, have been recently successful in modelling the turbulent properties of different regions of the inner heliosphere, i.e., the Earth's magnetosheath [37] and the near-Sun solar wind [38]. In particular, Franci et al. [38] have recovered the shape of the power spectrum of the magnetic field as observed by PSP during its first perihelion, with a transition region around the ion scales characterised by a spectral index compatible with $-11/3$. This is steeper than the values that are typically obtained at 1 au, which exhibit a statistical distribution peaking around $-2.8$, e.g., [39], although there is a tendency towards steeper power laws when the level of turbulent fluctuations is larger and/or the ion plasma beta $\beta_i$, i.e., the ratio of the ion thermal pressure to the magnetic pressure, is smaller, e.g., [40,41]. A similar trend is observed in numerical simulations with varying ion plasma beta [42], where the spectral index is found to be compatible with $-11/3$ for very small values of $\beta_i$ and increases with increasing $\beta_i$, being $\sim -3$ for $\beta_i \sim 1$, which is a typical average value for the solar wind at 1 au, e.g., [39,43,44].

Here, we analyse and compare two high-resolution 2D hybrid kinetic simulations of fully developed plasma turbulence covering two decades of scales around the ion characteristic scales. These have been previously found to reproduce well the spectral properties of solar wind turbulence at 0.17 and 1 au from the Sun. The former has been presented and compared to PSP observations in [38] and employs PSP-driven plasma parameters: it has a larger initial level of turbulent fluctuations with respect to the ambient magnetic field and $\beta_i < 1$ and exhibits a magnetic field power-law power spectrum with a spectral index of $-11/3$ at sub-ion scales. The latter has been analysed in Franci et al. [45], with specific focus on the role of magnetic reconnection in triggering a turbulent cascade at sub-ion scales: it has $\beta_i = 1$ and a smaller level of fluctuations and exhibits a less steep magnetic field spectrum with an ion-scale spectral index of $-3$.

Our comparative study allows us to identify and discuss many similarities and differences between the two simulations, while also gaining new insights on the development and nature of turbulence and magnetic reconnection under different plasma conditions. These include the fact that the magnetic and density fluctuations at sub-ion scales are mainly governed by the force balance equation, that strong elongated Alfvénic structures and magnetic field reversals—reminiscent of switchbacks—can form due to interacting magnetic vortices with aligned magnetic field lines, and that the onset and development of turbulence-mediated magnetic reconnection is determined by the eddy turnover time associated with the energy injection scale of the turbulence.

The paper is organized as follows. In Section 2, we describe the numerical dataset, providing details on the simulation setup and on the main plasma parameters. In Section 3, we present our results, analysing and comparing different types of structures (Section 3.1), their spectral properties (Section 3.2), and the interplay between turbulence and magnetic reconnection (Section 3.3). Finally, we discuss our findings in Section 4 and draw our conclusions in Section 5.

## 2. Methods

The two numerical simulations were performed using the hybrid particle-in-cell code CAMELIA [46,47], which solves the Vlasov–Maxwell equations, comprising the equations

of motion for individual ions, and the electron fluid equations. More specifically, it retains the kinetic physics associated with ions by modelling them as macroparticles—i.e., portions of their velocity distribution function—while it does not include any kinetic effect due to electrons, as these are treated as a massless, isothermal, charge-neutralising fluid.

The initial condition comprises a 2D homogeneous plasma in the $(x, y)$ plane with an out-of-plane uniform ambient magnetic field, $\boldsymbol{B}_0 = B_0 \hat{\boldsymbol{z}}$. This is initially perturbed with Alfvénic-like magnetic and ion bulk velocity fluctuations with only perpendicular components with respect to $\boldsymbol{B}_0$. These are the superposition of Fourier modes with equal amplitude, random phases, and energy equipartition between magnetic and kinetic energy and are characterised by a negligible initial cross-helicity.

Code units are the magnitude of the ambient field, $B_0$, for magnetic field fluctuations $\delta \boldsymbol{B}$; the Alfvén velocity, $V_A$, for the ion bulk velocity fluctuations $\delta \boldsymbol{u}_i$; the inverse ion gyrofrequency, $\Omega_i^{-1}$, for time, and the ion inertial length, $d_i = v_A / \Omega_i$, for space.

The only parameter that the two simulations have in common is the size of the simulation box, which is $256 \, d_i \times 256 \, d_i$. The main numerical and physical parameters are listed in Table 1. Run 1 resembles the conditions of the plasma environment met by PSP at its first perihelion, as the ion and electron plasma betas are set to values comparable to their average observed values, i.e., $\beta_i = 0.2$ and $\beta_e = 0.5$. Run 2 is instead more representative of the solar wind plasma at 1 au, where the typical values of both $\beta_i$ and $\beta_e$ are clustered around 1. Other physical parameters characterising the initial conditions for Run 1 and Run 2, respectively, are the maximum wavenumber of the initial magnetic field spectrum $k_\perp^{\text{inj}} d_i \lesssim 0.4$ and 0.3 (we refer to this quantity as the injection scale, and it is important as it also sets a limit for the turbulence correlation length) and the amplitude of the initial magnetic fluctuations $B^{\text{rms}} / B_0 \sim 0.44$ and $B^{\text{rms}} / B_0 \sim 0.25$, where $\Psi^{\text{rms}} = (\langle \Psi^2 \rangle - \langle \Psi \rangle^2)^{1/2}$ denotes the root mean square value (rms) of the quantity $\Psi$.

The numerical setting for Run 1 and Run 2 consists of, respectively, $4096^2$ and $2048^2$ grid points, a spatial resolution $\Delta x = \Delta y = d_i / 16$ and $d_i / 8$, 1024 and 64,000 particles-per-cell (ppc), and a resistivity $\eta = 1.5 \times 10^{-3}$ and $\eta = 0.5 \times 10^{-3}$ in units of $4\pi / \omega_i$. Further details about our numerical setup and its implementation can be found in Franci et al. [20], while more detailed information on the numerical and physical parameters and further analysis for Run 1 and Run 2 can be found in Franci et al. [38] and Franci et al. [45], respectively.

The two simulations are here analysed and compared at the time $t_{max}$, when their respective rms value of the current density, $J^{\text{rms}}$, reaches its maximum. This is considered as a reliable proxy that indicates when a quasi-stationary turbulent state has fully developed. As a further confirmation, we have verified that the spectral properties of different fields do not change significantly around $t_{max}$. We found $t_{max} = 50 \, \Omega_i^{-1}$ and $200 \, \Omega_i^{-1}$ for Run 1 and Run 2, respectively. The time evolution of some global quantities (including $J^{\text{rms}}$) for Run 2 has already been presented in Figure 1 of [45]. For Run 1, the time evolution is qualitatively the same but faster, in agreement with the fact that the nonlinear eddy turnover time $t_{\text{nl}}$ associated with the injection scales is smaller.

**Table 1.** List of simulations and their relevant numerical and physical parameters. From left to right, we report the number of points of the 2D box, the spatial resolution, $\Delta x$, the number of particles-per-cell, ppc, the resistivity, $\eta$, the rms amplitude of the initial magnetic fluctuations, $B^{\text{rms}}$, the injection scale, $k_\perp^{\text{inj}}$, the ion plasma beta, $\beta_i$, and the electron plasma beta, $\beta_e$. Finally, we list the corresponding references where the two simulations were first presented, which contain complementary information and analysis.

| Run | Grid | $\Delta x$ | ppc | $\eta$ | $\delta B^{\text{rms}}$ | $k_\perp^{\text{inj}} d_i$ | $\beta_i$ | $\beta_e$ | Ref. |
|-----|------|-----------|-----|--------|------------------------|---------------------------|-----------|-----------|------|
| 1 | $4096^2$ | $d_i/16$ | 1024 | 0.0015 | $0.44 \, B_0$ | 0.42 | 0.2 | 0.5 | [38] |
| 2 | $2048^2$ | $d_i/8$ | 64,000 | 0.0005 | $0.24 \, B_0$ | 0.28 | 1 | 1 | [45] |

## 3. Results

### 3.1. Different Types of Structures

Let us start our comparison between the two simulations by looking at how the magnetic and current structures appear when turbulence is fully developed and whether their shapes and spatial distribution differ. Figure 1 shows contour plots of the logarithm of the magnitude of the magnetic fluctuations, $|\delta B|^2 = |B - B_0|^2$ (left column), and of the current density, $|J|^2$ (right column), for Run 1 (top row) and Run 2 (bottom row). We have chosen to show logarithm values so that the small-scale structures can be better appreciated. In Figure 1a, we can observe the presence in Run 1 of many magnetic vortices with different radii, ranging from approximately $d_i$ up to approximately $10\,d_i$, which corresponds approximately to the injection scale. We also observe many elongated filamentary structures, with a width of the order of $d_i$ and length of a few tens of $d_i$. These typically form due to the interaction of two large-scale vortices, which become closer together and squeeze the magnetic field lines in between them, causing an increase in the magnetic field magnitude. If the magnetic field lines are directed in the same direction, magnetic reconnection via the formation of an X point does not take place. On the contrary, we have a region where the perpendicular components of the magnetic field are enhanced and this can extend for tens of $d_i$, following the edge between many vortices, until a region with oppositely directed magnetic field lines is encountered, and these will eventually break and reconnect. In these latter regions, we have the formation of strong current sheets, with a width of the order of $d_i$, as observed in Figure 1b. Current sheets keep forming due to interacting vortices and disrupting via reconnection during the whole evolution, and when a fully developed turbulent state is achieved, there is some form of balance in a statistical sense between these two processes, as we will see in more detail in Section 3.3. The magnetic structures in the two simulations, shown in Figure 1a–c, look qualitatively different. While, in Run 1, both vortices and filaments look quite smooth, in Run 2, they seem much more curly and rippled. The filling factor of such coherent structures also seems smaller than for Run 1. Finally, in Figure 1d, we observe many regions where the current is very small, especially inside the largest vortices, which does not seem to be the case in Figure 1b. The different qualitative behaviour is likely related to the total plasma beta: when this is smaller than 1, the magnetic pressure exceeds the particles' thermal pressure and it is therefore more difficult for the fluid to bend the magnetic field lines. It is also reasonable to expect an effect of the turbulence strength, leading to a different relative contribution from waves and coherent structures, with the former being more important when the level of turbulent fluctuations is smaller with respect to the ambient field. A detailed comparative spatiotemporal analysis, similar to that recently done in Papini et al. [33], would be required to further investigate the presence and contributions of different types of waves and will the subject of a future study.

We now analyse more quantitatively some of the different types of structures observed in the two simulations by extracting 1D data via cuts in the simulation domains. These are meant to resemble what a spacecraft would measure if it was flying through the box fast enough that modes and structures would not have the time to evolve with their characteristic time while being crossed. Under this hypothesis (frozen-in Taylor approximation), we can consider the 1D spatial series as if they were time series.

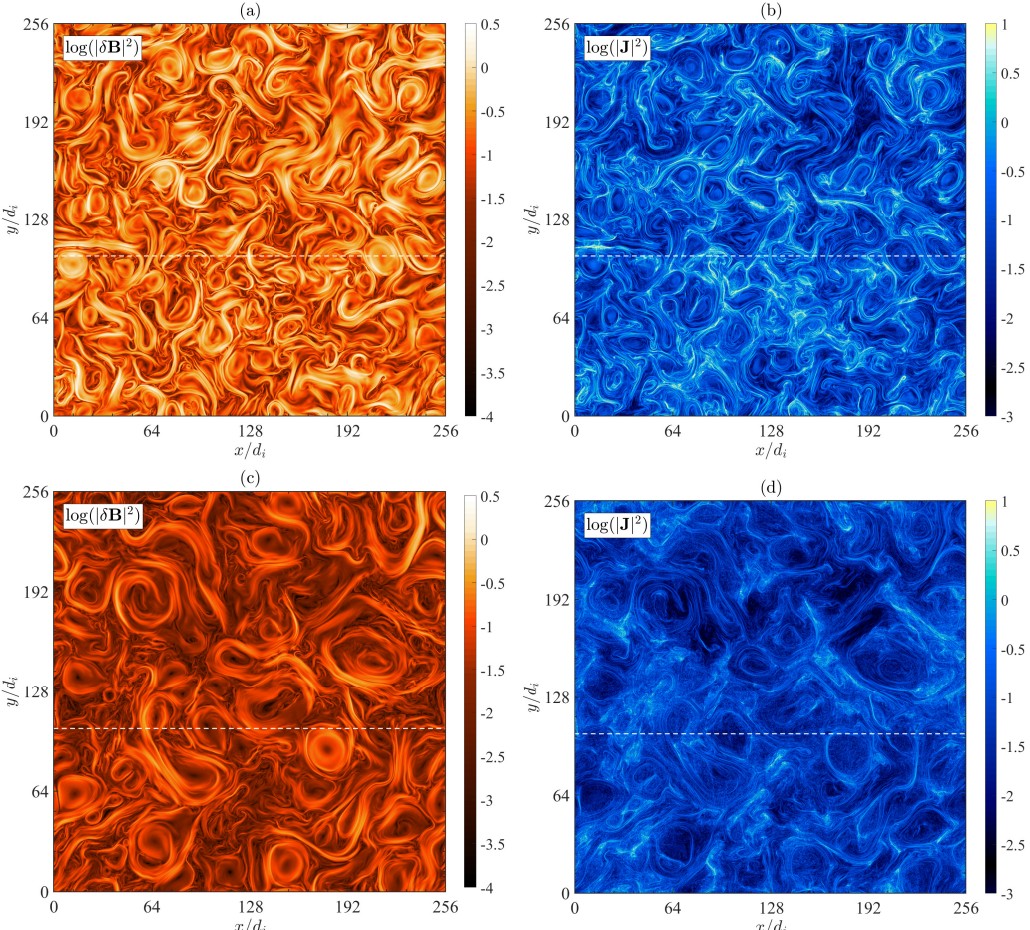

**Figure 1.** Contour plots of the logarithm of the magnitude of the magnetic fluctuations, $|\delta \mathbf{B}|^2 = |\mathbf{B} - \mathbf{B}_0|^2$, and of the current density, $|\mathbf{J}|^2$, for Run 1 ((**a**,**b**), respectively) and for Run 2 ((**c**,**d**), respectively), at their respective time of maximum turbulent activity. The white horizontal dotted lines mark 1D cuts that we analyse in Figures 2 and 3.

Figure 2 shows a horizontal 1D cut for Run 1 at $y = 104\, d_i$. We specifically chose this coordinate as it contains the absolute minimum value of $\mathbf{B}_z$, which even becomes negative. The different panels show 2D stripes around the cut for $|\delta \mathbf{B}|^2$ (a), $|\delta \mathbf{u}_i|^2$ (b), and $n$ (c), and the 1D cut for $|\mathbf{B}|$ and $n$ (d), the $x$ components of $|\delta \mathbf{B}|$ and $|\delta \mathbf{u}_i|$ (e), their $y$ components (f), and their $z$ component (g). In panels (e,f), we also show the ion bulk velocity components with an inverted sign in dashed lines, to highlight regions where the fluctuations are Alfvénic, by looking for $\delta \mathbf{B} \propto \pm \delta \mathbf{u}_i$. Finally, in panel (g), we also plot $\mathbf{B}_z = \mathbf{B}_0 + \delta \mathbf{B}_z$ to look for possible magnetic field reversals. Both $|\mathbf{B}|$ and $n$ exhibit quite large variations with respect to their average value, which is 1 (panel d). We also observe large variations in the perpendicular components of $\mathbf{B}$ and $\mathbf{u}_i$ (e,f), whose level in some regions is of the order of the ambient magnetic field. The variations of the $z$ components are quite small (g), except for a few small regions where they can locally overcome their perpendicular counterparts.

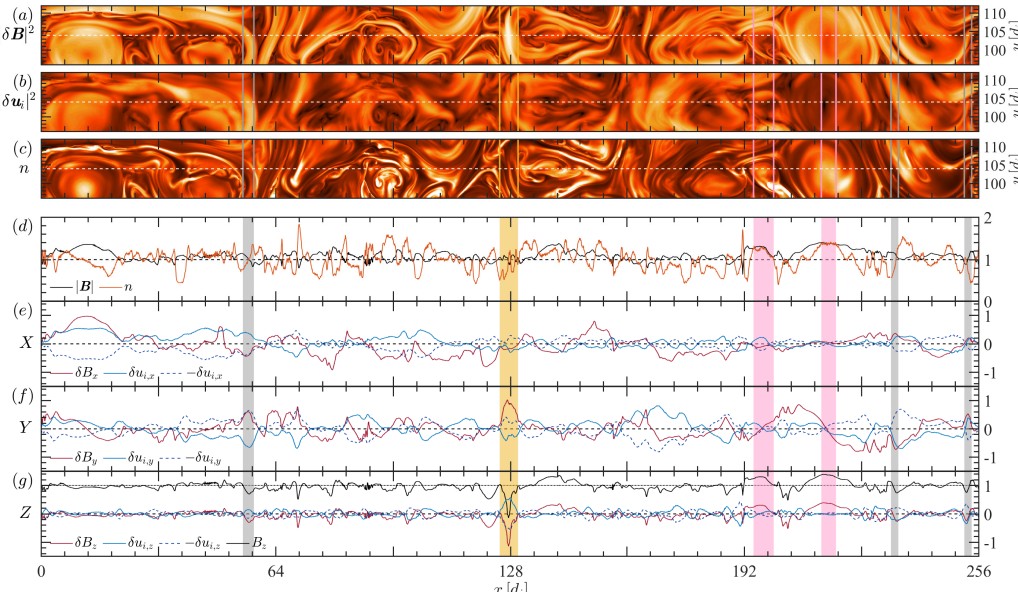

**Figure 2.** Horizontal 1D cut of the simulation domain for Run 1 at $y = 104\,d_i$. The three top panels show a 2D rectangular area of height $16\,d_i$ around the 1D cut for different fields: the magnitude of the magnetic fluctuations, $|\delta\boldsymbol{B}|^2$ (panel **a**), the magnitude of the ion bulk velocity fluctuations, $|\delta\boldsymbol{u}_i|^2$ (**b**), and the density, $n$ (**c**). The bottom four panels show the 1D cut for different scalar quantities, i.e., the magnitude of the magnetic field and the density (**d**), and the three components of the magnetic field and the ion bulk velocity (**e**–**g**). We compare each component $\delta B_{\hat{\jmath}}$ with both $\delta u_{i,\hat{\jmath}}$ and $-\delta u_{i,\hat{\jmath}}$ (with $\hat{\jmath} = x, y, z$) to better identify regions where the fluctuations are Alfvénic.

The three gray-shaded areas in panels (d)–(g) highlight regions where the fluctuations are highly Alfvénic, as $\delta B_{\hat{\jmath}} = \pm u_{i,\hat{\jmath}}$, with $\hat{\jmath} = x, y, z$. Looking back to panel (a), we can easily identify these as intense filamentary magnetic structures, with a width of the order of $d_i$, which have formed between large vortices. As we mentioned regarding Figure 1, these are regions where the magnetic field lines at the borders of interacting vortices have the same direction, so reconnection cannot occur. It is therefore reasonable to assume that, under these circumstances, the frozen-in condition holds, which is why, in these structures, we observe a strong coupling between magnetic field and ion bulk velocity fluctuations. In panel (d), we observe that the variations of $|\boldsymbol{B}|$ and $n$ occur on short length scales, of the order of the ion inertial length, and they typically seem anti-correlated, so that increases in the former correspond to decreases in the latter and vice versa, although there is not a perfect correspondence and the levels of the two are different, which can be linked to fact that the plasma beta is different from 1. It is reasonable to speculate that the anti-correlation between the two quantities is related to the balance between magnetic and particles' thermal pressure, so that when one decreases, the other one has to increase and vice versa. There are, however, two pink-shaded regions where we observe a direct correlation instead, as $\delta|\boldsymbol{B}| = \delta n$. These regions are as large as a few times $d_i$, and panel (a) shows that they correspond to the centres of large-scale vortices. In this case, the pressure balance cannot hold as the fluctuations of both quantities are positive with respect to their mean values. We can explain this by considering the Faraday–Ohm equation in the Hall-MHD approximation, which, when written for $\boldsymbol{B}/n$, reads $\boldsymbol{B}/n$, and reads $d(\boldsymbol{B}/n)/dt_e = (\boldsymbol{B}/n) \cdot \boldsymbol{\nabla}\boldsymbol{u}_e$, where $d(\boldsymbol{B}/n)/dt_e \equiv \partial(\boldsymbol{B}/n)/\partial t + \boldsymbol{u}_e \cdot \boldsymbol{\nabla}(\boldsymbol{B}/n)$ (cf. [48] for the ideal MHD case where $\boldsymbol{u}_e \to \boldsymbol{u}_i$).

Figure 1b shows that inside the two magnetic vortices corresponding to the pink-shaded areas, the amplitude of the current density is small and uniform, as the gradients of the magnetic field are mostly localised in its borders. The same can be said for the electron bulk velocity (not shown here), which provides the dominant contribution to the current. Moreover, in 2D, we have $(\boldsymbol{B}/n) \cdot \boldsymbol{\nabla}\boldsymbol{u}_e \equiv (\delta\boldsymbol{B}_\perp/n) \cdot \boldsymbol{\nabla}_\perp\boldsymbol{u}_e$, so that this term is always second-order in the fluctuations, with $\delta\boldsymbol{B}_\perp$ being small in these regions (see Figure 1c,d).

We can then assume $d(\boldsymbol{B}/n)/dt_e \simeq 0 \Rightarrow d(|\boldsymbol{B}|/n)/dt_e \simeq 0 \Rightarrow |\boldsymbol{B}|/n = c$, where $c$ is a constant: an increase in the magnitude of the magnetic field determines an increase in the density (and vice versa), in such a way that $\delta|\boldsymbol{B}| = c\,\delta n$.

Finally, it is worth discussing the yellow-shaded area, which appears quite peculiar. In this region, the fluctuation of the magnetic field in the $z$ direction is larger than $B_0$ and opposite in sign, resulting in a reversal of the local magnetic field. This resembles the so-called "switchbacks", i.e., structures that exhibit large rotations in the magnetic field direction and have been observed in the solar wind, especially closer to the Sun. Switchbacks are arc-polarized Alfvènic structures, e.g., [49–51], meaning that the magnetic field vector inside a switchback lies on the arc of a sphere of radius $|\boldsymbol{B}|$. They are also associated with proton velocity enhancements [52,53]. Here, we indeed recover some of these properties: (i) as already mentioned, the magnetic field direction is reversed (panel g); (ii) the fluctuations are predominantly Aflvènic, as we observe some overlapping between $\delta \boldsymbol{B}$ and $\pm \delta \boldsymbol{u}_i$ in all three components, except for some depletion of velocity fluctuations at the centre of the structure; (iii) the magnitude of the magnetic field, $|\boldsymbol{B}|$, is quite constant within the structure, in contrast with the larger variations that are observed in most other regions; (iv) the velocity component in the direction of $\boldsymbol{B}_0$ exhibits a large peak, of the order of the Aflvèn velocity, much larger than anywhere else in the 1D cut. It is worth stressing that here we are saying that the highlighted structure exhibits some "switchback-like" properties, with no claim that this can actually be regarded as one. Switchbacks are typically observed at much larger scales. In our simulation, however, the correlation length of the turbulence is limited by the size of the simulation box. We can speculate that in a much larger, realistic, 3D plasma system, where energy is injected by large-scale drivers such as large velocity shears, we could expect the formation of very large magnetic flux tubes; these could interact among each other and, if the magnetic field lines at their borders are directed in the same direction—as we can reasonably expect in the case of a strong ambient field such as that of the Parker spiral—the perpendicular magnetic field components could increase due to squeezing, and this could cause a very large decrease in the parallel component, as observed in our simulation. In such a situation, this kind of structure could form at much larger scales, which are not accessible here and/or grow in size also due to the solar wind expansion. In this regard, large-scale numerical simulations, also employing the expanding box model, could help to shed light on the formation and nature of switchbacks, as already suggested by Squire et al. [54].

Figure 3 shows a horizontal 1D cut for Run 2 at $y = 104\,d_i$. The panels here show the same quantities as for Run 1 in Figure 2. The fields exhibit a quite different behaviour with respect to Run 1: (i) the compressibility is much smaller, as the fluctuations of $n$ and $|\boldsymbol{B}|$ are much smaller than for Run 1 (panel d); (ii) there is a very good anti-correlation between $|\boldsymbol{B}|$ and $n$, with the fluctuations of the two quantities being comparable and opposite in sign almost everywhere along the 1D cut (d); (iii) there are many regions where the fluctuations are Alfvénic, with $\delta \boldsymbol{B} = \pm c\,\delta \boldsymbol{u}_i$ and $c$ of the order of 1 (e,f); (iv) the fluctuations of $\boldsymbol{B}$ and $\boldsymbol{u}_i$ in the direction of $\boldsymbol{B}_0$ are very small (g). Point (ii) is compatible with the prediction from the force balance equation[1], $\nabla P = \boldsymbol{J} \times \boldsymbol{B}$, where $P = P_i + P_e$ is the total thermal pressure (see also the Appendix in Papini et al. [33]). Using well-known vector identities, the equation above can be rewritten in the form

$$\nabla \left( P + \frac{|\boldsymbol{B}|^2}{2} \right) = (\boldsymbol{B} \cdot \nabla)\boldsymbol{B}. \tag{1}$$

When the turbulent fluctuations are sufficiently small with respect to the ambient magnetic field, we can neglect the second-order terms in the fluctuations on both sides of Equation (1). We then obtain

$$|\boldsymbol{B}| = \sqrt{B_0^2 \left( 1 + 2\frac{\delta B_\parallel}{B_0} + \frac{|\delta \boldsymbol{B}|^2}{B_0^2} \right)} \simeq B_0 \left( 1 + \frac{\delta B_\parallel}{B_0} \right), \tag{2}$$

which yields $\nabla(|\boldsymbol{B}|^2) = 2\,|\boldsymbol{B}|\,\nabla(|\boldsymbol{B}|) \simeq 2\,B_0\,\nabla(|\boldsymbol{B}|)$, and $(\boldsymbol{B}\cdot\nabla)\boldsymbol{B} \equiv (\delta\boldsymbol{B}_\perp\cdot\nabla_\perp)\delta\boldsymbol{B} \simeq 0$, where, in the latter, we have used the fact that the system is 2D. Assuming quasi-neutrality, $n_i = n_e = n$, we can express $P = nk(T_i + T_e) = nk(\beta_i + \beta_e)/2$, where $k_B$ is the Boltzmann constant. If we also assume that the gradients of the pressure are mainly due to gradients in the density, we can then rewrite Equation (1) as

$$\nabla\left(\frac{\beta_i + \beta_e}{2}n + B_0\,|\boldsymbol{B}|\right) \simeq 0, \tag{3}$$

which yields

$$\frac{\beta_i + \beta_e}{2\,B_0}\,\delta n \simeq -\delta|\boldsymbol{B}| + c, \tag{4}$$

where $c$ is a constant, which is smaller for a smaller level of turbulent fluctuation. For Run 2, where $\beta_i = \beta_e = 1$, this merely gives $\delta n \simeq -\delta|\boldsymbol{B}|$, as is indeed observed in Figure 3d. The fact that, in some regions, this does not hold exactly is likely due to the fact that the plasma is not perfectly isothermal and some small variations in the ion temperature occur locally. Moreover, the approximation of considering the fluctuations much smaller than $B_0$ might not be appropriate everywhere in the simulation box. In this regard, it is interesting to note that the approximation in Equation (2) is strictly valid only when

$$\frac{2B_0\delta B_\parallel}{|\delta\boldsymbol{B}|^2} \gg 1 \tag{5}$$

and, when this holds, Equation (2) also allows us to approximate $\delta|\boldsymbol{B}| \simeq \delta B_\parallel$. Indeed, comparing Figure 3d,g highlights the similar behaviour of $\delta|\boldsymbol{B}|$ and $\delta B_\parallel (= \delta B_z)$, although there is not a perfect correspondence everywhere.

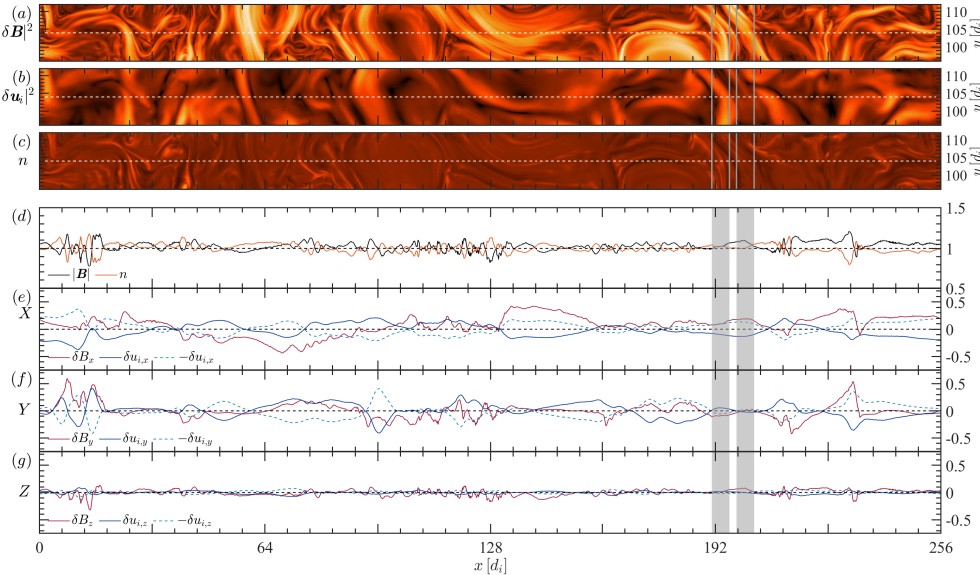

**Figure 3.** Same as for Run 1 in Figure 2, but for Run 2.

Finally, Figure 3 also shows that, even in Run 2, we can observe some regions where the fluctuations are Alfvénic (gray-shaded areas) in between vortices, although these seem to be less frequent than for Run 1 and slightly larger.

### 3.2. Spectral Properties

Now, we complement the description of the turbulent structures by analysing their spectral properties, as these are aspects that we can easily compare quantitatively between the two simulations. Figure 4 shows the power spectra of the magnetic and ion bulk velocity fluctuations for Run 1 (panel a) and for Run 2 (b), as well their direct comparison ((c) and (d)). In order to better compare the shape of the spectra, we have also renormalised the power spectra of $\boldsymbol{B}$ for Run 2 such that, at $k_\perp d_i = 1$, it has the same value as for Run 1. This normalisation factor corrects for the fact that the two simulations have two different initial levels of fluctuation and two slightly different injection scales. We have also renormalised the power spectrum of $\boldsymbol{u}_i$ for Run 2 by the same factor. The power spectra of $\boldsymbol{B}$ from both simulations exhibit a power-law behaviour with a spectral index compatible with $-5/3$ in the range between the injection scale and the ion scales. At smaller scales, it still behaves as a power law but with a different scaling, i.e., $-11/3$ for Run 1 (a) and $-3$ for Run 2 (b). Figure 4c shows that, once renormalised, the power spectra of $\boldsymbol{B}$ for the two simulations have exactly the same shape down to $k_\perp d_i \simeq 4$, before they diverge. The power spectra of $\boldsymbol{u}_i$ from both simulations also exhibit a power-law behaviour, with a spectral index that does not differ much from that of $\boldsymbol{B}$, although it has a lower level, which indicates the presence of some residual energy, i.e., an excess of magnetic over kinetic energy. At sub-ion scales, the power spectrum of $\boldsymbol{u}_i$ from Run 1 exhibits a power-law range with the same spectral index as $\boldsymbol{B}$ and a level that is less than an order of magnitude smaller, meaning that, at sub-ion scales, $\delta B_k$ and $\delta u_{i,k}$ are almost comparable until the noise level is reached. This hints at the possibility that, in this case, the ion bulk velocity fluctuations are somehow still coupled to the magnetic field fluctuations. On the contrary, the power spectrum of $\boldsymbol{u}_i$ from Run 2 drops at $k_\perp d_i \simeq 1$ and quickly falls a couple of orders of magnitude lower than the magnetic field spectrum. In this case, it seems that the ions fully decouple from the magnetic field once the ion kinetic scales are reached. Figure 4d confirms what has been said above, also clearly showing that the power spectrum of $\boldsymbol{u}_i$ for Run 2 starts decreasing at a larger scale with respect to Run 1. This could be related to the fact that the ion gyroradius $\rho_i = \sqrt{\beta_i} d_i$ is larger in the former, as is the ion beta. This trend is in agreement with what we previously observed in Franci et al. [42], where we analysed a collection of simulations with $\beta_i$ varying from 0.01 to 10 (although the power spectra of $\boldsymbol{u}_i$ were not explicitly shown there).

It is also interesting to compare the magnetic compressibility, $\delta B_\parallel^2 / |\delta \boldsymbol{B}|^2$, which is shown in Figure 5. In particular, in panel (a) and (b), we show the power spectra of the parallel magnetic fluctuations with respect to the guide field, $\delta B_\parallel$, of the magnitude of the magnetic field, $|\boldsymbol{B}|$, and of the density, $n$, for Run 1 and Run 2, respectively. Figure 5a shows that, for Run 1, the approximation $\delta B_\parallel \simeq \delta |\boldsymbol{B}|$ is quite good below the ion-scale break, while it is worse at larger scales. This is in agreement with the fact that the ratio in Equation (5) is minimum around the injection scale and it monotonically increases with decreasing scale, reaching a value of 10 at approximately the scale of the break (not shown here). In the inset, we show the ratio $\delta |\boldsymbol{B}| / \delta n$, which is constant for $k_\perp d_i \gtrsim 1$ and very close to the prediction from Equation (4), with $c$ negligible, which is marked by the black dashed horizontal line. It is worth noting that the ratio is constant only at sub-ion scales, while it is scale-dependent at larger scales. This could explain why we could not see a clear behaviour in the 1D cuts of the simulation domain shown in Figure 2. Figure 5b shows that, for Run 2, instead, the approximation $\delta B_\parallel \simeq \delta |\boldsymbol{B}|$ is very good for $k_\perp d_i \gtrsim 0.4$ and until the noise level is reached, in agreement with the fact that the ratio in Equation (5) reaches 10 already at $k_\perp d_i \simeq 1$. In this case, the fluctuations of $n$, $|\boldsymbol{B}|$, and $B_\parallel$ appear to be strongly coupled, since $(\beta_i + \beta_e)/2 = 1$, and such coupling holds at all scales above the noise level. This explains why we could observe a clear anti-correlation between $|\boldsymbol{B}|$ and $n$ in the 1D cuts shown in Figure 3. In Figure 5c, we directly compare the magnetic compressibility of the two simulations with the theoretical prediction $\delta B_\parallel^2 / |\delta \boldsymbol{B}|^2 = \beta / [2(1 + \beta)]$ [57,58], where $\beta = \beta_i + \beta_e$ is the total plasma beta, as we have also recently done in Matteini et al. [59]. Such prediction, which is valid for kinetic Alfvèn waves (KAWs), is expected to hold, more

in general, for low-frequency magnetic structures in pressure balance at those scales where the ion bulk velocity becomes negligible with respect to the electron one. Interestingly, the two simulations exhibit a comparable value of the magnetic compressibility at sub-ion scales, with a small shift in the scale at which a plateau is reached. Such shift is compatible with what we have found in Matteini et al. [59], where we have observed that such scale depends on the ion plasma beta, being smaller for larger betas. While the level is comparable to the theoretical prediction for Run 2, it is not for Run 1. It seems that, for both simulations, the level of magnetic compressibility sets to $\sim 1/3$, which corresponds to isotropy between the three magnetic field components. Finally, in Figure 5d, we compare the ratio $\delta B_\parallel^2 / \delta n^2$ for the two simulations with the theoretical prediction for KAWs. This can be rewritten as $\beta^2/4$ and it also corresponds to the prediction for magnetic structures in pressure balance in the presence of small turbulent fluctuations (cf. Equation (4)). While, for Run 2, the ratio almost coincides with the prediction, for Run 1, instead, we obtain an excess. We speculate that this could be related to the larger level of initial fluctuation, which affects the validity of our assumptions, as already discussed in regard to Figure 5a. It is important to note, however, that, for Run 1, the ratio is only less than a factor of 2 larger than the prediction, meaning that the ratio $\delta B_\parallel / \delta n$ is still of the order of 1.

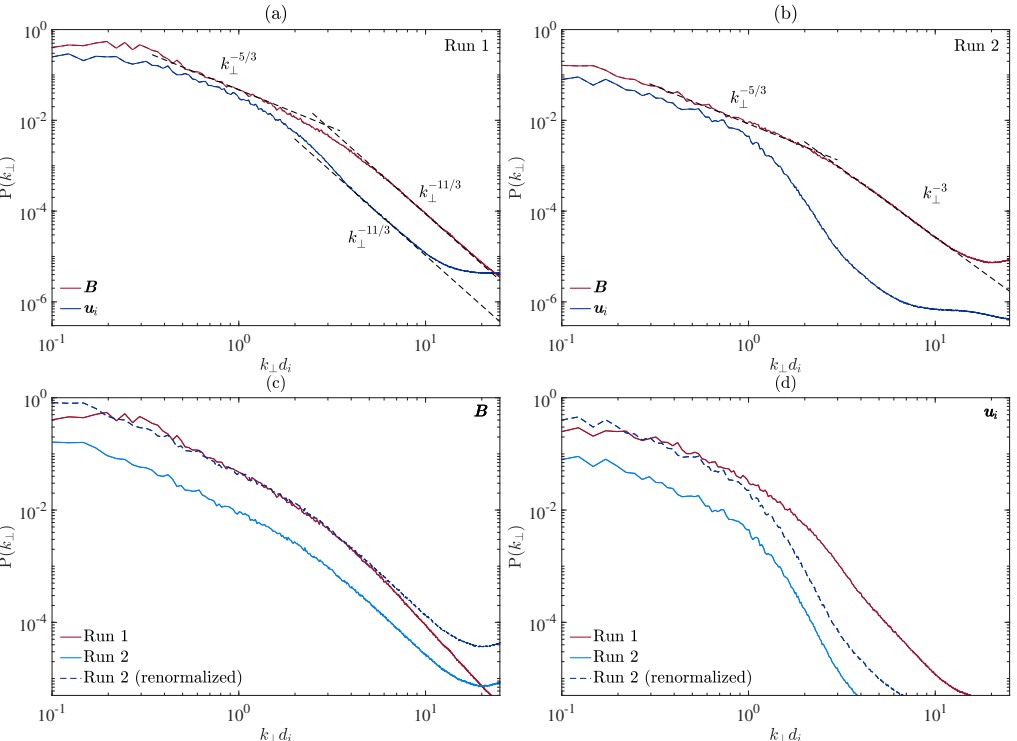

**Figure 4.** Spectral properties of the magnetic field and ion bulk velocity fluctuations: (**a**) power spectra of $\boldsymbol{B}$ and $\boldsymbol{u}_i$ for Run 1 with reference power laws superimposed; (**b**) same as panel (**a**) but for Run 2; (**c**) direct comparison of the magnetic field spectrum of Run 1 and Run 2, with the latter also renormalised by a factor that "corrects" for the lower initial magnetic fluctuations and the slightly larger injection scale (so that the spectra of the two simulations coincide at $k_\perp d_i = 1$); (**d**) same as panel (**c**) but for the ion bulk velocity.

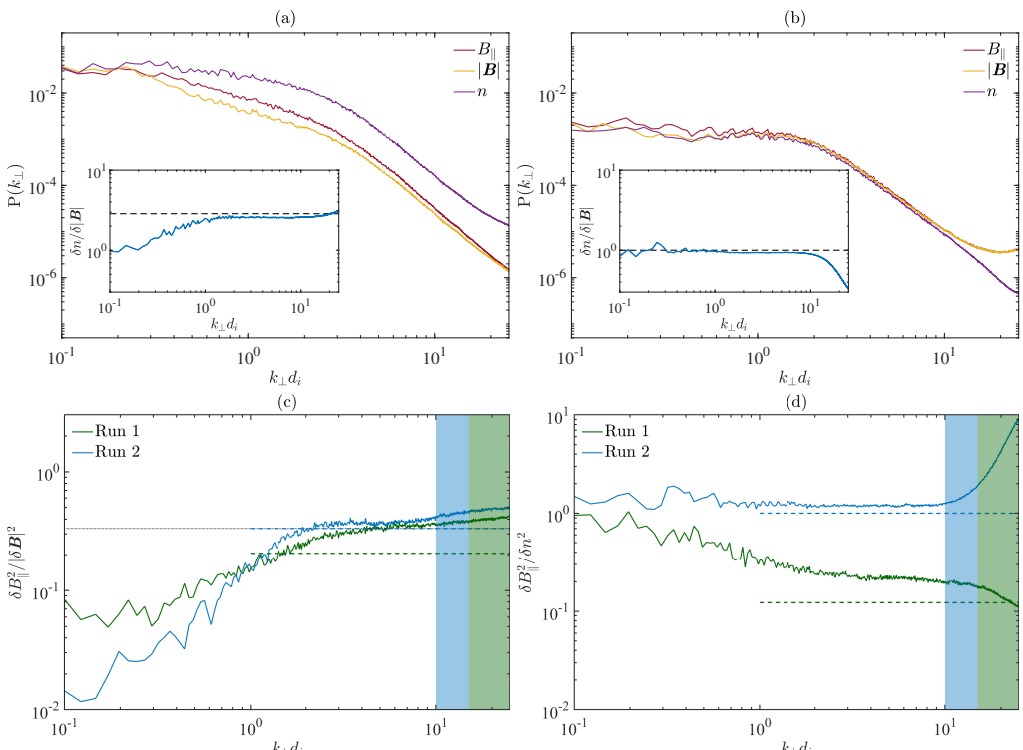

**Figure 5.** Spectral properties of the parallel magnetic fluctuations with respect to the ambient field, $B_\parallel$, and of the density, $n$: (**a**) power spectra of $B_\parallel$, $|\boldsymbol{B}|$, and $n$ for Run 1, with the ratio $\delta n / \delta |\boldsymbol{B}|$ in the inset; (**b**) same as for panel (**a**) but for Run 2; (**c**) magnetic compressibility $\delta B_\parallel^2 / |\delta \boldsymbol{B}|^2$ for Run 1 and Run 2, compared with theoretical predictions (see text); (**d**) ratio $\delta B_\parallel^2 / \delta n^2$ for Run 1 and Run 2, compared with theoretical predictions (see text). The shaded areas in panels (**c**,**d**) mark the scales at which the noise significantly affects the spectra in the two simulations.

### 3.3. Magnetic Reconnection

Magnetic reconnection is intrinsically linked to plasma turbulence. On the one hand, it occurs spontaneously in turbulent plasmas as the result of the interaction between nearby magnetic vortices, which create strong current sheets that eventually disrupt, e.g., [20]. On the other hand, it can act as a driver for the onset of a turbulent cascade below the ion characteristic scales, transferring energy non-locally in Fourier $k$-space from the inertial-range scales, characteristic of large magnetic vortices, to the sub-ion scales, characteristic of the current sheets [45]. It is therefore interesting to compare how magnetic reconnection behaves in the two simulations, under different plasma conditions. Figure 6 shows statistics of the evolution and properties of reconnection events. A detailed explanation of how these are detected and their reconnection rates are computed can be found in Papini et al. [60]. Here, we recall that we identify a reconnection event by pairing magnetic X-points with their nearest O-point inside a current sheet. For each of these events, we compute its reconnection rate as

$$\gamma_{\text{rec}} = \left| \frac{1}{\Phi|_X^O} \frac{\partial \Phi|_X^O}{\partial t} \right|, \tag{6}$$

where $\Phi|_X^O = A_z^O - A_z^X$ is the reconnected magnetic flux density between the O-point and the X-point, i.e., the difference in the out-of-plane vector potential, $A_z$, at the two points.

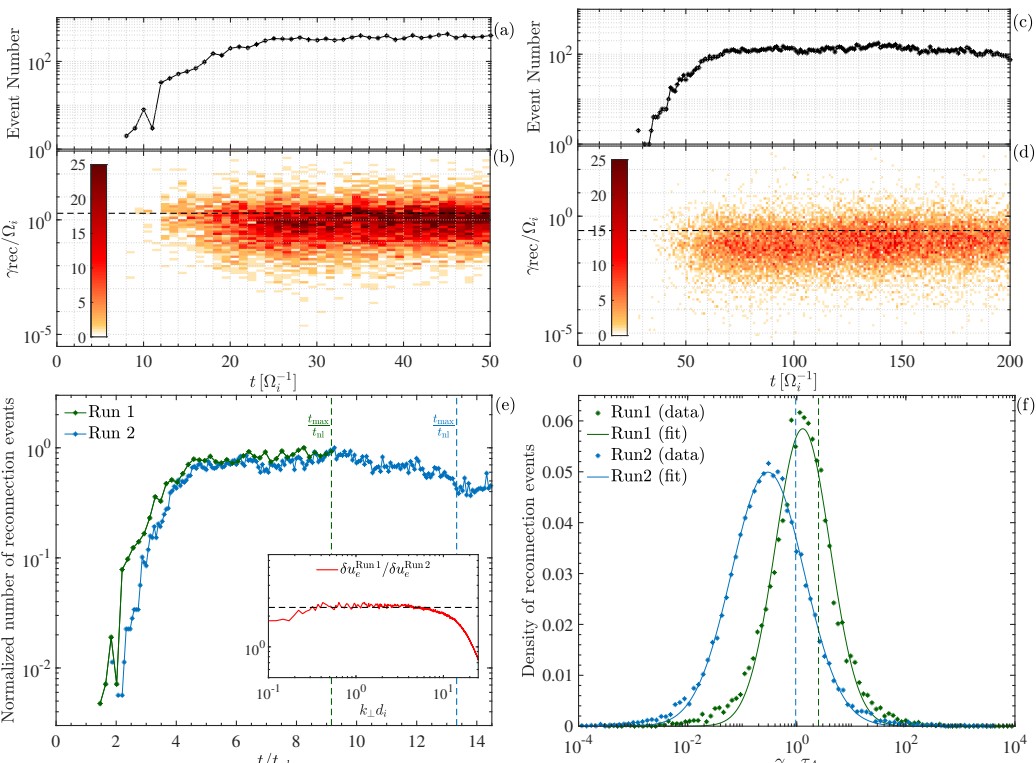

**Figure 6.** Overview of the time evolution of the properties of magnetic reconnection events: (**a**,**b**) number of reconnection events in the simulation box and their distribution in terms of the reconnection rate, $\gamma_{\mathrm{rec}}$, at each time during the evolution for Run 1 (the vertical dashed line marks the average reconnection rate); (**c**,**d**) same as panels (**a**,**b**) but for Run 2; (**e**) number of reconnection events normalised to their maximum values versus the time normalised to the nonlinear time for the two simulations; (**f**) distribution of reconnection rates at the time of maximum turbulent activity for the two simulations, normalised to their local Alfvén time $\tau_A$ (the vertical dashed lines mark the average values).

Figure 6a,c show the time evolution of the number of reconnection events during Run 1 and Run 2, respectively, up to the time when they reach a fully developed turbulent state. The overall evolution in both simulations is similar and qualitatively the same as the one reported in [60]. Firstly, we note that the number of reconnection events is different in the two simulations. This is a consequence of the different injection scale and of the different plasma conditions, which, as we have noticed in Section 3.1, determine the different filling factors of magnetic vortices. This leads to a different number of regions where vortices interact and hence to a different number of formed current sheets. Magnetic reconnection starts to occur early in both simulations, with the number of reconnection events growing until it reaches a more or less constant plateau, i.e., a balance is achieved between current sheets disrupting and new ones forming. Figure 6b,d show the distribution of the reconnection rates for all events at all times. The distribution of the reconnection rates for Run 1 is shifted toward higher values than for Run 2 by roughly one order of magnitude, with an average reconnection rate $\gamma_{\mathrm{rec}}/\Omega_i = 1.93$ and $0.34$ for Run 1 and Run 2, respectively. Therefore, it is reasonable to ask whether this is the indication of the existence of two different regimes of turbulence-mediated reconnection in the two runs or it is instead the same regime but occurring under different turbulence conditions. To further investigate this aspect, in Figure 6e, we compare the temporal evolution of the number of reconnection events in the two simulations, by putting together panels (a) and (c). To correct for the different number of events in the two simulations, here, we have renormalised the number of events by their respective maximum values during the evolution. Moreover, since we know that magnetic reconnection takes place in current sheets that form and shrink between magnetic vortices and that the first current sheets form

between the largest vortices, we have also renormalised the evolution time by the nonlinear time of the large-scale magnetic vortices (the so-called eddy turnover time), as this is the characteristic timescale of turbulence regardless of the different initial conditions that may be set, e.g., [61]. For the nonlinear time, we adopt the definition $t_{\mathrm{nl}} = (\delta u_e^{\mathrm{inj}} k_\perp^{\mathrm{inj}})^{-1}$, where $\delta u_e^{\mathrm{inj}}$ is the amplitude of the electron bulk velocity fluctuations at the injection scale. This yields $t_{\mathrm{nl}}^{\mathrm{Run}\,1} = 5.5\,\Omega_i^{-1}$ and $t_{\mathrm{nl}}^{\mathrm{Run}\,2} = 15\,\Omega_i^{-1}$. The time evolution in units of $t/t_{\mathrm{nl}}$ is very similar for the two simulations: they both develop the first reconnection events after approximately $\sim 1 t_{\mathrm{nl}}$ and reach the maximum turbulent activity after a few times this value. This is consistent with our previous results [45,60,62], and further confirms a posteriori that the driver of reconnection is the turbulence, via the interaction of magnetic vortices, and so the eddy turnover time also acts as the "characteristic clock" for the formation of current sheets, which then undergo reconnection.

Indeed, Figure 6e suggests the idea that the characteristic scale of magnetic reconnection mainly depends on some global parameters ruled by the turbulence. Therefore, we can expect also the average reconnection rate of the distribution of reconnection events to display a general dependence on these parameters. We can find such a dependence under the hypothesis that the tearing mode theory can be applied here, thanks to the large aspect ratio of the current sheets generated by turbulence. We can then identify the average reconnection rate with the growth rate of the average dominant reconnecting mode. We recall that, in order for tearing mode theory to be applicable, the growth rate must be much larger than the inverse eddy turnover time governing the evolution of the current sheet. Although this is strictly valid only in the case of a periodic current sheet, in the absence of periodicity, one may in principle assume the Fourier mode decomposition required by the tearing analysis to be valid in a Wentzel–Kramers–Brillouin (WKB) sense, if the current sheet aspect ratio is large enough (see Section IX of Betar et al. [63] for a discussion of these arguments applied to a case of turbulence-mediated reconnection).

The classic tearing mode theory provides specific scaling for the reconnection rate of the fastest growing mode [64]. This is indeed true in the case of current sheets with a sufficiently large aspect ratio, as has been verified in several numerical studies of magnetic reconnection [65–67] and turbulence-induced reconnection [45,60]. While the scalings of the fastest mode are only weakly dependent on the profile of the reconnecting current sheet [68], they display a more important dependence on non-ideal terms [63,68]. Luckily, regardless of the tearing mode regime and of the theoretical model considered, the growth rates of tearing-type modes are of the order of the local Alfvèn time $\tau_A \sim a/c_A$ when the non-ideal normalised parameters (i.e., $ac_A/\eta$, $(d_e/a)^2$, $(\rho_s/a)^2$, $(\rho_i/a)^2$, $(d_i/a)$, etc.) are of the order of $\sim 0.1$ or larger (e.g., see Table 2 of [69]). Here, $a$ is the characteristic width of the current sheet, $c_A$ is the local Alfvén speed measured at the current sheet location, $d_e$ is the electron inertial length, and $\rho_s$ is the sound gyroradius. Therefore, we can expect that the majority of the reconnection events of each simulation have a growth rate of the order of $\gamma_{\mathrm{rec}} \tau_A \sim \mathcal{O}(1)$. To obtain an estimate of $\gamma_{\mathrm{rec}}$, we then need to estimate the local Alfvén time of the current sheets in the simulation. Firstly, we observe that reconnection events occur in current sheets generated by turbulence, which are all of comparable size. We assume that the characteristic width $a$ corresponds to the scale of the ion-scale spectral break in the power spectrum of the magnetic fluctuations, as suggested in Franci et al. [42]. This assumption is based on the fact that such a break coincides with the maximum of the power spectrum of the current density, which we expect to occur at the scale corresponding to the width of the most intense current structures: current sheets are observed to be squeezed by the magnetic vortices, growing in amplitude, until they eventually disrupt so that, below this scale, the amplitude of the current fluctuations drops. We can then estimate $a$ by using the empirical formula for the ion-scale break from [42]. This was obtained from simulations which employed $\beta_i = \beta_e$. This condition is true for Run 2 and is also approximatively satisfied by Run 1. We recall that this formula agrees, in the limits of small and large $\beta_i$, with the solar wind data analysed by [70], which respectively yield $a \sim d_i$ and $a \sim \rho_i$. These limit values also agree with the theoretical estimates—obtained by dimensional analysis of the fluid equations

extended to include the full pressure tensor dynamics—of the characteristic spatial scales of current and vorticity structures, when steady conditions are assumed. These estimates yield $a \sim d_i$ for $\beta_e, \beta_i \ll 1$ and $a \sim \sqrt{\rho_s^2 + \rho_i^2}$ for $\beta_e, \beta_i \gg 1$ [71]. By applying the formula in Equation (1) of [42], we then obtain $a \simeq 0.37\,d_i$ for Run 1 and $a \simeq 0.50s\,d_i$ for Run 2. For what concerns the estimate of the Alfvén speed, we can assume that it is proportional to the initial average level of the magnetic fluctuations at the injection scale, i.e., $c_A \sim \delta B_{k^{\mathrm{inj}}}$. Since $\mathrm{P}_B(k^{\mathrm{inj}}) = \delta B_{k^{\mathrm{inj}}}^2/k^{\mathrm{inj}} \sim (\delta B^{\mathrm{rms}})^2$, it follows that $c_A \sim B^{\mathrm{rms}}\sqrt{k_{\mathrm{inj}}}$. By using the estimated values for $a$ and $c_A$, we then obtain $\tau_A = 1.30\,\Omega_i^{-1}$ and $3.94\,\Omega_i^{-1}$ for Run 1 and Run 2, respectively.

Figure 6f shows the distribution of the reconnection rates, normalised with respect to the average local Alfvén time $\tau_A$, at the time of maximum turbulent activity for the two simulations (i.e., a 1D cut from panel (b) at $t = 50\,\Omega_i^{-1}$ and from panel (d) at $200\,\Omega_i^{-1}$). The use of $\tau_A$ as a normalisation factor brings the two distributions very close to each other, with the normalised average reconnection rates being $\gamma_{\mathrm{rec}}\tau_A = 2.50$ for Run 1 and 0.96 for Run 2 (see the vertical dashed lines), which are consistent with the theoretical estimate, given the simplifying hypotheses made.

## 4. Discussion

The comparisons between Run 1 ($\beta_i = 0.2$, $\beta_e = 0.5$, $\delta B^{\mathrm{rms}} = 0.44\,B_0$) and Run 2 ($\beta_i = \beta_e = 1$, $\delta B^{\mathrm{rms}} = 0.24\,B_0$) have highlighted some similarities—some of which could not be given a priori, or at least not fully—as well as some significant differences. The common features are as follows:

- Magnetic structures of different shape and size form, e.g., vortices with a diameter ranging from the injection scales down to fractions of the inertial length $d_i$ and filamentary structures of width comparable to a few times $d_i$ and length up to a few tens of $d_i$. The latter form between vortices when their respective field lines are aligned and some of them are observed to be highly Alfvénic.
- Ion-scale current sheets form as the result of the interaction between vortices with anti-aligned magnetic field lines and they reconnect quite shortly afterwards. During the evolution, current sheets keep forming and disrupting via reconnection and a balance is soon reached between these two processes.
- The width of current sheets when they disrupt is comparable in the two simulations and of the order of the ion scales, likely due to the small difference in the ion plasma beta, such that $\rho_i = \sqrt{\beta_i}\,d_i = 0.45\,d_i$ for Run 1 and $\rho_i = d_i$ for Run 2.
- The power spectrum of the magnetic field exhibits a power-law behaviour with a spectral index of $-5/3$ in the inertial range, followed by an ion-scale spectral break.
- The power spectrum of the ion bulk velocity has a slightly lower level than the magnetic field one (indicative of residual energy) and it starts dropping at a scale which is larger than the break scale for the magnetic field.
- $\delta|\boldsymbol{B}| \simeq \delta B_\parallel$ represents a good approximation at sub-ion scales and the ratio $\delta n / \delta|\boldsymbol{B}|$ is perfectly constant and comparable to the theoretical prediction for magnetic structures in force balance.
- The magnetic compressibility increases from the inertial range through the ion scales until it reaches a plateau around the break scale of the magnetic field power spectrum, at a value that is comparable in the two simulations and corresponds to equipartition between the three vector components.

The main differences between the two simulations are, instead, as follows.

- The magnetic structures (vortices and filaments) appear well defined and with neat borders in Run 1, while they appear more curly and wave-like in Run 2.
- The filling factor of magnetic vortices is larger in Run 1, which also determines a larger number of reconnection events at each time during the evolution.
- $\delta|\boldsymbol{B}|$ and of $\delta n$ are large for Run 1, while they are much smaller for Run 2, where the two quantities exhibit a remarkable anti-correlation. This is directly related to

$\delta n / \delta |\boldsymbol{B}|$ being perfectly constant in Fourier space for $k_\perp d_i \simeq 1$ and very close to the expectation for structures in force balance (which also coincides with the KAW theoretical prediction) for Run 2 but less so for Run 1.

- Regions of correlation between $|\boldsymbol{B}|$ and $n$ are observed in Run 1, while they are not present in Run 2.
- The approximation $\delta |\boldsymbol{B}| \simeq \delta B_\parallel$ holds in the inertial range for Run 2, due to the smaller turbulent fluctuations, while it does not for Run 1.
- The power spectrum of the magnetic field is around an order of magnitude larger for Run 1, due to the larger initial level of fluctuations with respect to the ambient field.
- The power spectrum of the ion velocity at sub-ion scale falls quickly several orders of magnitude lower than the magnetic field one, with no well-defined behaviour, for Run 1, while, for Run 2, it exhibits a power law with the same spectral index as the magnetic field and a level that is less than an order of magnetic lower.
- The power spectrum of the density in Run 1 is around an order of magnitude larger than the magnetic field one, while, in Run 2, the two are almost overlapped.

Some of these differences in fact become similarities, once some key quantities are renormalised properly and complementary analysis is performed.

- The power spectra of the magnetic field of the two simulations, renormalised to take into account the different injections of energy at the top of the turbulent cascade, exhibit exactly the same shape at scales larger than the ion-scale break, as they overlap almost perfectly for $k_\perp d_i \simeq 4$.
- The very good anti-correlation between $n$ and $|\boldsymbol{B}|$ and the poor one observed in the 1D spatial cuts of Run 1 and Run 2, respectively, seems to suggest a different nature of the fluctuations. The ratio $\delta n / \delta |\boldsymbol{B}|$ in Fourier space, however, shows that the difference is only in the inertial range, where the ratio between the power spectra of the two quantities is constant in Run 2 but not in Run 1. At sub-ion scales, instead, the fluctuations are compatible with structures in force balance in both cases, when taking into account their respective plasma beta.
- The time evolution of magnetic reconnection is similar in the two simulations, once the simulation time is renormalised to the eddy turnover time of the injection-scale vortices, which is characteristic of the nonlinear time at the top of the turbulent cascade, $t_{\mathrm{nl}}$: reconnection events start to occur already at $t \simeq t_{\mathrm{nl}}$, and at approximately twice this time, they reach their maximum number and maintain it during the following evolution, when a balance between current sheet formation and disruption is achieved.
- The distributions of the reconnection rates of all magnetic reconnection events in the two simulations are comparable once we renormalise them by their respective Alfvèn time $\tau_A$, with merely a factor of $\sim 2$ of difference between their average values. These are also compatible with $< \gamma_{\mathrm{rec}} > \tau_A \sim \mathcal{O}(1)$, where $\tau_A$ depends on the rms of the magnetic fluctuations, on the energy injection scale, and on the ion plasma beta.

## 5. Conclusions

We have compared different properties of turbulence and magnetic reconnection in two high-resolution 2D hybrid simulations of turbulent plasmas with conditions typical of the solar wind close to the Sun (Run 1) and close to Earth (Run 2). The two simulations have both been initialised with in-plane Alfvénic-like magnetic and ion velocity fluctuations embedded in an out-of-plane ambient magnetic field. They differ, however, in some fundamental physical parameters, i.e., the ion and electron plasma beta ($\beta_i = 0.2$ and $\beta_e = 0.5$ for Run 1, $\beta_i = \beta_e = 1$ for Run 2) and the level of initial fluctuations with respect to the ambient field ($\delta B^{\mathrm{rms}} = 0.25\, B_0$ for Run 1 and $\delta B^{\mathrm{rms}} = 0.44\, B_0$ for Run 2). The two simulations also differ in their numerical setting (e.g., number of grid points, spatial resolution, number of particles). We are confident, however, that these do not play a significant role in determining the main differences, based on a large collection of simulations with varying numerical parameters and convergence tests performed over many years, e.g., [20]. By looking at the 1D spatial cuts of different fields across the simulation domains, at the power

spectra of different fields and some characteristic ratios in Fourier space, and at the statistic of magnetic reconnection events, we have identified similarities and differences between the two simulations.

The main conclusion of this work is that some features and properties of turbulence and reconnection, which appear quite different at first glance, are actually the manifestation of similar processes under different plasma conditions, rather than the result of different regimes. Our analysis also provides some interesting insights on the development and properties of plasma turbulence and magnetic reconnection in collisionless plasmas:

(i)     Regardless of their nature in the inertial range, the magnetic and density fluctuations are observed to be governed by the force balance at sub-ion scales, where the level of the fluctuations becomes small enough with respect to the ambient field;

(ii)    Strong elongated Alfvénic structures can form between two vortices (or, likely, between two flux tubes in a more realistic 3D environment), when their respective field lines are aligned and magnetic reconnection does not occur;

(iii)   When the level of turbulent fluctuations is large enough, there is a chance to develop magnetic field reversals, with properties that are similar to those of switchbacks;

(iv)    When the ion plasma beta is small and/or the turbulence strength is large, which correspond to the conditions where the power spectrum of the magnetic field is typically steeper at sub-ion scales, e.g., [72], the power spectrum of the ion velocity seems to be not negligible, hinting at a possible role of the ion current in the turbulent cascade [38];

(v)     Under the plasma conditions mentioned above, the level of magnetic compressibility at sub-ion scales seems to be determined by the magnetic field component isotropy and, as such, to be independent of the plasma beta, rather than by the theoretical prediction for KAWs;

(vi)    When magnetic reconnection develops spontaneously due to the interaction of turbulent magnetic vortices, the eddy turnover time at the energy injection scale is the "clock" that governs the evolution of reconnection events; these start to develop after roughly one characteristic time, and after a few times this value, a balance is reached between the current sheets forming and those disrupting via reconnection. The characteristic time for their disruption via reconnection is instead compatible with the Alfvén time of the background turbulence.

It is important to stress that this numerical study suffers from some intrinsic limitations. First, the two simulations analysed here are 2D. Although our previous studies have suggested that the spectral properties do not change significantly between two and three dimensions [38,47], we cannot exclude that other turbulence properties and magnetic reconnection would not be substantially altered in 3D and that the two simulations would not differ more. For example, temperature anisotropy-driven instabilities are strongly affected by the dimensionality (cf. Hellinger et al. [73] vs. Hellinger et al. [74]). Secondly, the simulation box size limits the injection scale and, as a consequence, the maximum sizes of magnetic vortices, current sheets, and switchback-like structures. Last but not least, the level of realism of the simulations could be much further improved by also setting the initial ion temperature anisotropy to observed average values (here, we start with temperature isotropy) and including the effects of the solar wind expansion. Extending this comparison by using large-scale 3D expanding-box hybrid simulations, also complementing previous studies that investigated the effects of the solar wind expansion, e.g., [74–77], will be the subject of future work.

**Author Contributions:** Conceptualisation, L.F., E.P. and D.D.S.; Methodology, L.F., P.H. and E.P.; Software, P.H. and L.F.; Validation, L.F., E.P. and D.D.S.; Formal analysis, L.F. and E.P.; Investigation, L.F., E.P. and D.D.S.; Resources, L.F. and D.B.; Data curation, L.F.; Writing—original draft preparation, L.F.; Writing—review and editing, L.F., E.P., D.D.S., P.H., D.B., L.M., S.L. and V.M.-C.; Visualization, L.F.; Funding acquisition, D.B. and L.F. All authors have read and agreed to the published version of the manuscript.

**Funding:** L.F. and D.B. are supported by the UK Science and Technology Facilities Council (STFC), grant ST/T00018X/1. This work was performed using the DiRAC Data Intensive Service at Cambridge, which is operated by the University of Cambridge Research Computing service as part of the Cambridge Service for Data Driven Discovery (CSD3), and the DiRAC Data Intensive service at Leicester (DIaL), operated by the University of Leicester IT Services, both of which form part of the STFC DiRAC HPC Facility (www.dirac.ac.uk). The DiRAC component of CSD3 was funded by BEIS capital funding via STFC capital grants ST/P002307/1 and ST/R002452/1 and STFC operations grant ST/R00689X/1. The DIaL equipment was funded by BEIS capital funding via STFC capital grants ST/K000373/1 and ST/R002363/1 and STFC DiRAC Operations grant ST/R001014/1. DiRAC is part of the National e-Infrastructure. Access to DiRAC resources was granted through Director's Discretionary Time allocations in 2019 and 2020. We acknowledge CINECA for the availability of high-performance computing resources and support under the program Accordo Quadro MoU INAF-CINECA "Nuove frontiere in Astrofisica: HPC e Data Exploration di nuova generazione" (project "INA20 C6A55"). We acknowledge PRACE for awarding us access to SuperMUC-NG at GCS@LRZ, Germany, and Marconi at CINECA, Italy. The discussion of the results of this work has been facilitated by the Project HPC-EUROPA3 (INFRAIA-2016-1-730897), with the support of the EC Research Innovation Action under the H2020 Programme (grants HPC177WO5I and HPC17MTH1N). We acknowledge funding by Fondazione Cassa di Risparmio di Firenze under the project "HIPER-CRHEL".

**Institutional Review Board Statement:** Not applicable.

**Informed Consent Statement:** Not applicable.

**Data Availability Statement:** Not applicable.

**Acknowledgments:** The authors acknowledge useful discussions with Julia Stawarz and the members of the Solar Orbiter Science Working Groups.

**Conflicts of Interest:** The authors declare no conflict of interest. The funders had no role in the design of the study, in the collection, analyses, or interpretation of data; in the writing of the manuscript, or in the decision to publish the results.

## Note

[1] Note that here we have assumed that the ion pressure tensor can be approximated as a scalar quantity, so that $\nabla \cdot \boldsymbol{P} \simeq \nabla P$, although ion pressure anisotropy is predicted to be generated in Alfvénic turbulence at $k_\perp d_i \simeq 1$ [55,56]. This approximation is good enough for providing an interpretation of the observed correlation between magnetic and density fluctuations.

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
