# Peer review of "Plasma Turbulence in the Near-Sun and Near-Earth Solar Wind: A Comparison via Observation-Driven 2D Hybrid Simulations"

_universe, doi:10.3390/universe8090453_

Round 1

Reviewer 1 Report

In this work, the Authors compared the two simulation results which represent the near-Sun and the near-Earth solar wind. These two simulations are already present in the authors' previous articles. The simulation results at the time when the current density reaches its maximum for the two cases are chosen for the comparison, and the authors draw the conclusion that some features and properties of solar wind turbulence and reconnection, differ from each other at a first glance, are the manifestation of similar regimes under different plasma conditions. 

I consider the topic of solar wind plasma turbulence simulation to be very important, and the authors give sufficient simulation data, theoretical estimations, and discussion.

However, some minor issues should be fixed before accepting for publication.

Here are my detailed comments.

Minor points:

1)  In Line 101, change "to the their average" to "to their average", please confirm.

2)  In Line 108 and the following text, "Brms" and "Jrms" should be defined, or a brief description and a ref should be given.

3)  In Figure 1, the contour plots are the logarithm of corresponding values, but in the top left corner of each subplot, the name strings do not have "log", which could confuse readers. 

    Also, the reasons for showing logarithm values are suggested to be given.

4)  In Figure 2(a), 2(b), 2(c), 3(a), 3(b), 3(c), are these plots all logarithm values too? the value names on the left side are suggested to include "log" function.

5)  In Line 148-150, the authors conclude that magnetic structures are more well-defined in Run 1 than in Run 2.  As I see, the grid size(delta x) in Run 1 is smaller than in Run 2, this may contribute to this effect.

6)  In Line 168, the perturbation of Bz can exceed B0z, which results in a negative value of Bz. The authors do not give a description of the simulation algorithm as it is listed in the authors' previous articles, which I have not read, but I suppose the algorithm equations are calculated with the small perturbation of physical values, so, will the large perturbation of Bz lead to unreliable of simulation results or not?

7)  In Page 9, the label of Figures 4(c) and 4(d) on the top of subplots are wrong, please correct them.

8)  In Line 369, the full name of WKB, Wentzel-Kramers-Brillouin, is suggested to  give. 

9)  In Line 394, change "We can them estimate" to "We can then estimate", please confirm.

10) In Line 464, change "less than an oder of" to "less than an order of", please confirm.

11) In Line 501-503, have you discussed the effects of numerical setting in previous articles?   or are there any refs? 

Reviewer 2 Report

The paper desciribes application of the numerical model CAMELIA for 2D hybrid simulation of plasma turbulence with observation-driven initial conditions which are representative of the near-Sun and the near-Earth solar wind. It includes a comparison of simulations in two cases: the near-Sun case at 0.17 au taking the fudamental parameters measured by the Parker Solar Probe and the near-Earth position at 1 au with the typically observed average parameters. This comparison demonstrates a link between spatial and spectral properties of the magnetic, ion velocity, and plasma density fluctuations with development of turbulence. It was found that in both cases some physical phenomena are similar. In particular, it concerns formation of the ion-scale Alfvenic structures, specific orientation of the magnetic fild lines in absence of magnetic reconnection, origination of the magnetic and density fluctuations at sub-ion scales,the characteristic time of the formation of current sheets, etc.  The model used for simulations is well documented, all parameters,  calculations and results are detaily commented. The references are numerous and sufficient, the style of the text is clear. In general, the paper seems quite convincing. I recommend to publish it the Special Issue: Advances in Solar Wind Origin and Evolution of Universe after minor corrections of misprints (e.g. "flucuations"at the lines 52, 514) and style of the references.

Author Response

The would really like to thank the reviewer for appreciating our manuscript and considering it convincing, suggesting it for publication.

We have corrected the suggested typos and others throughout the manuscript and fixed some issues with the references.